# On the Parameterization of Gaussian Mean Field Posteriors in Bayesian Neural Networks

## Abstract

Variational Bayesian Inference is a popular methodology for approximating posterior distributions in Bayesian neural networks. Recent work developing this class of methods has explored ever richer parameterizations of the approximate posterior in the hope of improving performance. In contrast, here we share a curious experimental finding that suggests instead restricting the variational distribution to a more compact parameterization. For a variety of deep Bayesian neural networks trained using Gaussian mean-field variational inference, we find that the posterior standard deviations consistently exhibit strong low-rank structure after convergence. This means that by decomposing these variational parameters into a low-rank factorization, we can make our variational approximation more compact without decreasing the models' performance. Furthermore, we find that such factorized parameterizations improve the signal-to-noise ratio of stochastic gradient estimates of the variational lower bound, resulting in faster convergence.

## 1 Introduction

**Bayesian Neural Networks** explicitly represent their parameter-uncertainty by forming a *posterior distribution* over model parameters, instead of relying on a single point estimate for making predictions, as is done in traditional deep learning. For neural network weights $\mathbf{w}$, features $\mathbf{x}$ and labels $\mathbf{y}$, the posterior distribution $p(\mathbf{w}|\mathbf{x}, \mathbf{y})$ is computed using Bayes' rule, which multiplies the prior distribution $p(\mathbf{w})$ and data likelihood $p(\mathbf{y}|\mathbf{w}, \mathbf{x})$ and renormalizes. When predicting with Bayesian neural networks, we form an average over model predictions where each prediction is generated using a set of parameters that is randomly sampled from the posterior distribution. Bayesian neural networks are thus a type of *ensembling*, of which various types have proven highly effective in deep learning (see e.g. Goodfellow et al., 2016, sec 7.11).

Besides offering improved predictive performance over single models, Bayesian ensembles are also more robust because ensemble members will tend to make different predictions on hard examples (Raftery et al., 2005). In addition, the diversity of the ensemble represents predictive uncertainty and can be used for out-of-domain detection or other risk-sensitive applications (Ovadia et al., 2019).

**Variational inference** is a popular class of methods for approximating the posterior distribution $p(\mathbf{w}|\mathbf{x}, \mathbf{y})$, since the exact Bayes' rule is often intractable to compute for models of practical interest. This class of methods specifies a distribution $q_{\boldsymbol{\theta}}(\mathbf{w})$ of given parametric or functional form as the posterior approximation, and optimizes the approximation by solving an optimization problem. In particular, we minimize the Kullback-Leibler (KL) divergence $D_{KL}$ between the variational distribution $q_{\boldsymbol{\theta}}(\mathbf{w})$ and the true posterior distribution $p(\mathbf{w}|\mathbf{x}, \mathbf{y})$, which is given by

$$D_{\mathrm{KL}}[q_{\boldsymbol{\theta}}(\mathbf{w})||p(\mathbf{w}|\mathbf{x}, \mathbf{y})] = \mathbb{E}_q\left[\log \frac{q_{\boldsymbol{\theta}}(\mathbf{w})}{p(\mathbf{w}|\mathbf{x}, \mathbf{y})}\right] = \mathbb{E}_q\left[\log \frac{q_{\boldsymbol{\theta}}(\mathbf{w})}{p(\mathbf{w})p(\mathbf{y}|\mathbf{w}, \mathbf{x})/p(\mathbf{y}|\mathbf{x})}\right].$$

Here, we do not know the normalizing constant of the exact posterior $p(\mathbf{y}|\mathbf{x})$, but since this term does not depend on $\mathbf{w}$, we may ignore it for the purpose of optimizing our approximation $q$. We are then left with what is called the negative Evidence Lower Bound (negative ELBO):

$$L_q = D_{\mathrm{KL}}[q_{\boldsymbol{\theta}}(\mathbf{w})||p(\mathbf{w})] - \mathbb{E}_q[p(\mathbf{y}|\mathbf{w}, \mathbf{x})].$$

In practice, the expectation of the likelihood $p(\mathbf{y}|\mathbf{w}, \mathbf{x})$ with respect to $q$ is usually not analytically tractable and instead is estimated using Monte Carlo sampling:

$$\mathbb{E}_q[\log p(\mathbf{y}|\mathbf{w}, \mathbf{x})] \approx \frac{1}{S}\sum_{s=1}^{S}\log p(\mathbf{y}|\mathbf{w}^{(s)}, \mathbf{x}), \quad \mathbf{w}^{(s)} \sim q_{\boldsymbol{\theta}}(\mathbf{w}),$$

where the ELBO can then be optimized by differentiating this stochastic approximation with respect to the variational parameters $\boldsymbol{\theta}$ (Salimans et al., 2013; Kingma & Welling, 2013).

In **Gaussian Mean Field Variational Inference** (GMFVI), we choose the variational approximation to be a fully factorized Gaussian distribution $q = \mathcal{N}(\boldsymbol{\mu}_q, \boldsymbol{\Sigma}_q)$ with $w_{lij} \sim \mathcal{N}(\mu_{lij}, \sigma_{lij}^2)$, where $l$ is a layer number, and $i$ and $j$ are the row and column indices in the layer's weight matrix. Although this type of approximation is considered to be one of the simplest types of variational approximations, it already doubles the parameter count compared to deterministic neural networks. In addition, Bayesian neural networks with mean-field Gaussian posterior approximations often are harder to train than deterministic neural networks because they suffer from increased noise in stochastic gradient estimates.

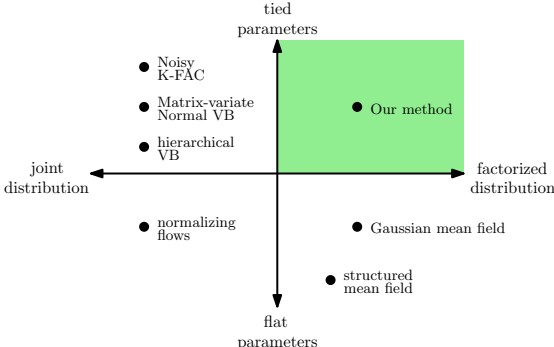

**Figure 1:** Approaches to variational Bayes on Bayesian neural networks, ordered by i) whether they factorize the variational distribution $q$, and ii) whether they tie the variational parameters.

Beyond mean-field variational inference, recent work on approximate Bayesian inference has explored ever richer parameterizations of the approximate posterior in the hope of improving the performance of Bayesian neural networks (see Figure 1). In contrast, here we study a simpler, more compactly parameterized variational approximation, which we show can also work well for a variety of models. In particular we find that:

- Converged posterior standard deviations under GMFVI consistently display strong low-rank structure. This means that by decomposing these variational parameters into a low-rank factorization, we can make our variational approximation more compact without decreasing our model's performance.
- Factorized parameterizations of posterior standard deviations are easier to train since they improve the signal-to-noise ratio of stochastic gradient estimates of the variational lower bound. Using a more limited parameterization of the approximate posterior can thus lead to faster convergence and a reduction in number of parameters compared to standard GMFVI.

## 2  MEAN FIELD POSTERIOR STANDARD DEVIATIONS NATURALLY HAVE LOW-RANK STRUCTURE

In this section we show that the converged posterior standard deviations of Bayesian neural networks trained using standard GMFVI consistently display strong low-rank structure. We also show that it is possible to compress the learned posterior standard deviation matrix using a low-rank approximation without decreasing the network's performance. We first briefly introduce the mathematical notation for our GMFVI setting and the low-rank approximation that we explore. We then provide experimental results that support the two main claims of this section.

To avoid any confusion among the readers, we would like to clarify that we use the terminology "low-rank" in a particular context. While variational inference typically makes use of low-rank decompositions to compactly represent the *dense covariance* of a Gaussian variational distribution (see numerous references in Section 4), we investigate instead underlying low-rank structures within *the already diagonal covariance* of a Gaussian fully-factorized variational distribution. We will make this explanation more formal in the next section.

## 2.1 METHODOLOGY

To introduce the notation we consider layers that consist of a linear transformation followed by a non-linearity $f$,

$$\mathbf{a}_l = \mathbf{h}_l \mathbf{W}_l + \mathbf{b}_l, \qquad \mathbf{h}_{l+1} = f(\mathbf{a}_l), \tag{1}$$

where $\mathbf{W}_l \in \mathbb{R}^{m \times n}$, $\mathbf{h}_l \in \mathbb{R}^{1 \times m}$ and $\mathbf{b}_l, \mathbf{a}_l, \mathbf{h}_{l+1} \in \mathbb{R}^{1 \times n}$. To simplify the notation in the following, we drop the subscript $l$ such that $\mathbf{W} = \mathbf{W}_l$, $\boldsymbol{\mu}_q = \boldsymbol{\mu}_{ql}$, $\boldsymbol{\Sigma}_q = \boldsymbol{\Sigma}_{ql}$ and we focus on the kernel matrix $\mathbf{W}$ for a single layer.

In GMFVI, (Blundell et al., 2015), we model the variational posterior as

$$q(\mathbf{W}) = \mathcal{N}(\boldsymbol{\mu}_q, \boldsymbol{\Sigma}_q) = \prod_{i=1}^{m} \prod_{j=1}^{n} q(w_{ij}), \quad \text{with} \quad q(w_{ij}) = \mathcal{N}(\mu_{ij}, \sigma_{ij}^2), \tag{2}$$

where $\boldsymbol{\mu}_q \in \mathbb{R}^{mn \times 1}$ is the posterior mean vector, $\boldsymbol{\Sigma}_q \in \mathbb{R}_+^{mn \times mn}$ is the diagonal posterior co-variance matrix. The weights are then usually sampled using a reparametrization trick (Kingma & Welling, 2013), i.e, for the $s$-th sample, we have

$$w_{ij}^{(s)} = \mu_{ij} + \sigma_{ij} \epsilon^{(s)}, \qquad \epsilon \sim \mathcal{N}(0, 1). \tag{3}$$

In practice, we often represent the posterior standard deviation parameters $\sigma_{ij}$ in the form of a matrix $\mathbf{A} \in \mathbb{R}_+^{m \times n}$. Note that we have the relationship $\boldsymbol{\Sigma}_q = \text{diag}(\text{vec}(\mathbf{A}^2))$ where the elementwise-squared $\mathbf{A}$ is vectorized by stacking its columns, and then expanded as a diagonal matrix into $\mathbb{R}_+^{mn \times mn}$.

In the sequel, we start by empirically studying the properties of the spectrum of matrices $\mathbf{A}$ *post training*, while using standard Gaussian mean-field variational distributions. Interestingly, we observe that those matrices naturally exhibit a low-rank structure (see Section 2.3 for the corresponding experiments), i.e,

$$\mathbf{A} \approx \mathbf{U}\mathbf{V}^T \tag{4}$$

for some $\mathbf{U} \in \mathbb{R}^{m \times k}$, $\mathbf{V} \in \mathbb{R}^{n \times k}$ and $k$ a small value (e.g., 2 or 3). This observation motivates the introduction of the following variational family, which we name $k$-tied Normal:

$$k\text{-}tied\text{-}\mathcal{N}(\mathbf{W}; \boldsymbol{\mu}_q, \mathbf{U}, \mathbf{V}) = \mathcal{N}\big(\boldsymbol{\mu}_q, \text{diag}\big(\text{vec}((\mathbf{U}\mathbf{V}^T)^2)\big)\big), \tag{5}$$

where the squaring of the matrix $\mathbf{U}\mathbf{V}^T$ is applied elementwise. Due to the tied parametrization of the diagonal covariance matrix, we emphasize that this variational family is *smaller*—i.e., included in—the standard Gaussian mean-field variational distribution family.

As formally discussed in Appendix B, the matrix variate Gaussian distribution (Gupta & Nagar, 2018), referred to as $\mathcal{MN}$ and already used for variational inference by Louizos & Welling (2016) and Sun et al. (2017), is related to our $k$-tied Normal distribution with $k = 1$ when $\mathcal{MN}$ uses diagonal row and column covariances. Interestingly, we prove that for $k \geq 2$, our $k$-tied Normal distribution cannot be represented by any $\mathcal{MN}$ distribution. This illustrates the main difference of our approach from

| Variational family | Parameters (total) |
|---|---|
| Multivariate Normal | $mn + \frac{mn\,(mn+1)}{2}$ |
| diagonal Normal | $mn + mn$ |
| $\mathcal{MN}$ | $mn + \frac{m(m+1)}{2} + \frac{n(n+1)}{2}$ |
| $\mathcal{MN}$(diagonal) | $mn + m + n$ |
| $k$-tied Normal | $mn + k(m + n)$ |

**Table 1:** Number of variational parameters for a variational family for a matrix $\mathbf{W} \in \mathbb{R}^{m \times n}$. $\mathcal{MN}$(diagonal) is from Louizos & Welling (2016).

the most closely related previous work of Louizos & Welling (2016) (see also Figure 1). Furthermore, notice that diagonal covariance $\boldsymbol{\Sigma}_q$ repeatedly reuses the same elements of $\mathbf{U}$ and $\mathbf{V}$, which results in parameter sharing across different weights.

The total number of the standard deviation parameters in our method is $k(m + n)$ from $\mathbf{U}$ and $\mathbf{V}$, compared to $mn$ in the standard GMFVI parametrization from $\mathbf{A}$. Given that in our experiments the $k$ is very low (e.g. $k = 2$) this reduces the number of parameters from quadratic to linear in the dimensions of the layer, see Table 1. More importantly, such parameter sharing across the weights leads to higher signal-to-noise ratio during training and thus faster convergence. We demonstrate this phenomena in the next section. In the rest of this section, we will first demonstrate that the standard GMFVI methods already learn a low-rank structure in the posterior standard deviation matrix $\mathbf{A}$. Furthermore, we will show that replacing the full matrix $\mathbf{A}$ with its low-rank approximation does not reduce the predictive performance.

## 2.2 EXPERIMENTAL SETTING

Before providing the experimental results we briefly explain the key properties of the experimental setting. We analyse three types of GMFVI Bayesian neural network models:

- Multilayer Perceptron (MLP): a network of 3 dense layers and ReLu activations that we train on the MNIST dataset (LeCun & Cortes, 2010). We use the last 10,000 examples of the training set as a validation set.

- Convolutional Neural Network (CNN): a LeNet architecture (LeCun et al., 1998) with 2 convolutional layers and 2 dense layers that we train on the CIFAR-100 dataset (Krizhevsky et al., 2009b). We use the last 10,000 examples of the training set as a validation set.

- Long Short-Term Memory (LSTM): a model that consists of an embedding and an LSTM cell (Hochreiter & Schmidhuber, 1997), followed by a single unit dense layer. We train it on an IMBD dataset (Maas et al., 2011), in which we use the last 5,000 examples of the training set as a validation set.

- Residual Convolutional Neural Network (ResNet): a ResNet-18[1] architecture (He et al., 2016) trained on all 50,000 training examples of the CIFAR-10 dataset (Krizhevsky et al., 2009a).

In each of the four models we use a mean-field Normal posterior and a Normal prior with a single scalar standard deviation hyper-parameter for all the layers. We optimize the variational parameters using an Adam optimizer (Kingma & Ba (2014)). For a more comprehensive explanation of the experimental setup used in this section please refer to Appendix A.1. Finally, we highlight that our experiments focus primarily on the comparison across a broad range of model types rather than competing with the state-of-the-art results over the specifically used datasets. Therefore, we use mainly small to medium models (MLP, CNN, LSTM) that are known to train well using the standard GMFVI approach explored in this paper. Encouraged by the reviewers, we also briefly show that our results can extend to larger models such as the ResNet-18 model. However, scaling GMFVI to such larger model sizes is still a challenging research problem (Osawa et al., 2019).

## 2.3 MAIN EXPERIMENTAL OBSERVATION

Our main experimental observation is that the standard GMFVI learns posterior standard deviation matrices that have a low-rank structure across different model types (MLP, CNN, LSTM, ResNet) and layer types (dense and convolutional). To show this, we investigate the results of the SVD decomposition of posterior standard deviation matrices for the four types of models trained until ELBO convergence using GMFVI. While we evaluate the low-rank structure only for the dense layers of the first three models (MLP, CNN and LSTM), we investigate also the low-rank structure of the convolutional layers of the ResNet model.

Figure 2 shows the percentage of explained variance per singular value $k$ of the SVD decomposition of dense layers in the first three models. The percent of explained variance for the singular value $k$ is calculated as $\gamma_k^2 / \sum_{i'} \gamma_{i'}$, where $\gamma_{i'}$ are singular values. We observe that most of the variance in the posterior standard deviation parameters is captured in the rank-1 approximation. However, a more fine-grained analysis shows that a rank-2 approximation can encompass nearly all of the remaining variance. Finally, we note that we do not observe the same behaviour for the posterior mean parameters as we do for the posterior standard deviation parameters. Figure 9 in Appendix D further supports this claim visually by comparing the heat maps of the full-rank posterior standard deviations matrix with its rank-1 and rank-2 approximations. In particular, we observe that the rank-2 approximation results in the heat-map looking visually very similar to the full-rank matrix. Finally, Figure 3 illustrates that the low-rank structure is also visible in both the dense and convolutional layers of the ResNet model. See Appendix C for more details. In the analysis of the experiments, we use the shorthand SEM to refer to the standard error of the mean.

## 2.4 LOW-RANK APPROXIMATION OF VARIANCE MATRICES

---

[1]`https://github.com/tensorflow/probability/blob/master/tensorflow_`
`probability/examples/cifar10_bnn.py`

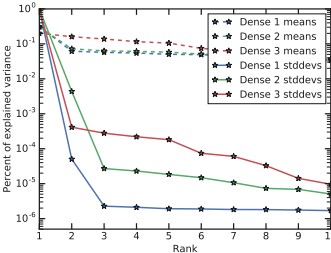 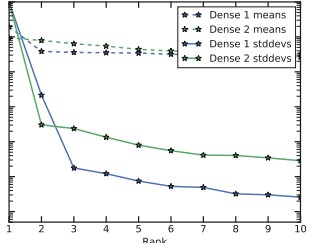 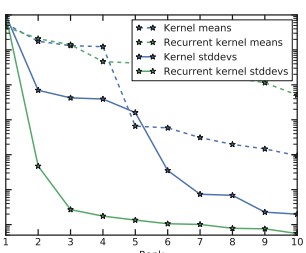

**Figure 2:** Explained variance per singular value from SVD of matrices of posterior means and posterior standard deviations for different layers of three types of models trained using standard GMFVI: MLP (left), CNN (center), LSTM (right). Posterior standard deviations clearly display strong low-rank structure, with most of the variance contained in the top few singular values, while this is not the case for posterior means.

| | MLP | | | CNN | | | LSTM | | |
|---|---|---|---|---|---|---|---|---|---|
| Rank | -ELBO ↓ | NLL ↓ | Accuracy ↑ | -ELBO ↓ | NLL ↓ | Accuracy ↑ | -ELBO ↓ | NLL ↓ | Accuracy ↑ |
| Full | $0.431_{\pm0.0057}$ | $0.100_{\pm0.0034}$ | $97.6_{\pm0.15}$ | $3.83_{\pm0.020}$ | $2.23_{\pm0.017}$ | $42.1_{\pm0.49}$ | $0.536_{\pm0.0058}$ | $0.493_{\pm0.0057}$ | $80.1_{\pm0.25}$ |
| 1 | $3.41_{\pm0.019}$ | $0.677_{\pm0.0040}$ | $93.6_{\pm0.25}$ | $4.33_{\pm0.021}$ | $2.30_{\pm0.016}$ | $41.7_{\pm0.49}$ | $0.687_{\pm0.0058}$ | $0.491_{\pm0.0056}$ | $80.0_{\pm0.25}$ |
| 2 | $0.456_{\pm0.0059}$ | $0.107_{\pm0.0033}$ | $97.6_{\pm0.15}$ | $3.88_{\pm0.020}$ | $2.24_{\pm0.017}$ | $42.2_{\pm0.49}$ | $0.621_{\pm0.0058}$ | $0.494_{\pm0.0057}$ | $80.1_{\pm0.25}$ |
| 3 | $0.450_{\pm0.0059}$ | $0.106_{\pm0.0033}$ | $97.6_{\pm0.15}$ | $3.86_{\pm0.020}$ | $2.24_{\pm0.017}$ | $42.1_{\pm0.49}$ | $0.595_{\pm0.0058}$ | $0.493_{\pm0.0056}$ | $80.1_{\pm0.25}$ |

**Table 2:** Impact of low-rank approximation of the GMFVI-trained posterior standard deviation matrix on ELBO and predictive performance, for three types of models. We report mean and SEM of each metric across 100 models samples.

Motivated by the above observation, we show that it is possible to replace the full-rank posterior standard deviation matrix with its low-rank approximation without a decrease in performance. Table 2 shows the comparison of performance of models with different ranks of approximation to their posterior standard deviation matrix for the MLP, CNN and LSTM models. Figure 3 contains analogous results for the ResNet model. The results show that the post-training approximation with ranks higher than 1 achieves predictive performance close to that of the full-rank matrix for all the analyzed model and layer types. This observation itself could be used as a form of post-training network compression. Moreover, it gives rise to further interesting exploration directions such as formulating posteriors that exploit such a low rank structure. In the next section we explore this particular direction while focusing on the first three model types (MLP, CNN, LSTM).

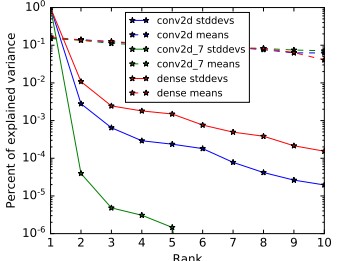

| Rank | -ELBO ↓ | NLL ↓ | Accuracy ↑ |
|---|---|---|---|
| Full | $122.61_{\pm0.012}$ | $0.495_{\pm0.0080}$ | $83.5_{\pm0.37}$ |
| 1 | $122.57_{\pm0.012}$ | $0.658_{\pm0.0069}$ | $81.7_{\pm0.39}$ |
| 2 | $122.77_{\pm0.012}$ | $0.503_{\pm0.0080}$ | $83.2_{\pm0.37}$ |
| 3 | $122.67_{\pm0.012}$ | $0.501_{\pm0.0079}$ | $83.2_{\pm0.37}$ |

**Figure 3:** Unlike posterior means, the posterior standard deviations of both dense and convolutional layers in the ResNet model trained using standard GMFVI display strong low-rank structure and can be approximated without loss in predictive metrics. Top: Explained variance per singular value of the matrices of converged posterior means and standard deviations. Bottom: Impact of *post training* low-rank approximation of the posterior standard deviation matrices on model's performance. We report mean and SEM of each metric across 100 models samples.

## 3 THE $k$-TIED NORMAL DISTRIBUTION: EXPLOITING LOW-RANK PARAMETER-STRUCTURE IN MEAN FIELD POSTERIORS

In the previous section we have shown that it is possible to replace a full-rank matrix of posterior standard deviations trained with GMFVI with this matrix's low-rank approximation without decreasing the predictive performance. In this section we show that it is also possible to exploit this observation during training time. We achieve this by exploiting our novel variational family, the $k$-tied Normal distribution (see Section 2.1).

We show that using this distribution in the context of GMFVI in Bayesian neural networks allows to reduce the number of network parameters and in some cases speed up model convergences while maintaining the predictive performance of the standard parametrization of the GMFVI.

We start by recalling the definition of the $k$-tied Normal distribution:

$$k\text{-}tied\text{-}\mathcal{N}(\mathbf{W}; \boldsymbol{\mu}_q, \mathbf{U}, \mathbf{V}) = \mathcal{N}\big(\boldsymbol{\mu}_q, \mathrm{diag}\big(\mathrm{vec}\big((\mathbf{U}\mathbf{V}^T)^2\big)\big)\big)$$

where the variational parameters are comprised of $\{\boldsymbol{\mu}_q, \mathbf{U}, \mathbf{V}\}$.

## 3.1 EXPERIMENTAL SETTING

We now introduce the experimental setting in which we evaluate the GMFVI variational posterior parametrized by the $k$-tied Normal distribution. We assess the impact of the described posterior in terms of predictive performance and reduction in the number of parameters for the same first three model types (MLP, CNN, LSTM) and respective datasets (MNIST, CIFAR-100, IMDB) as we used in the previous section. Additionally, we also analyse the impact of tying in the posterior on the signal-to-noise ratio of stochastic gradient estimates of the variational lower bound for the CNN model as a representative example. Overall the experimental setup is very similar to the one introduced in the previous section. Therefore, we highlight here only the key differences.

We apply the $k$-tied Normal variational posterior distribution to the layers which we analysed in the previous section. More concretely, we use the $k$-tied Normal variational posterior for all the three layers of the MLP model, the two dense layers of the CNN model and the LSTM cell's kernel and recurrent kernel. We initialize the parameters $u_{ik}$ and $v_{jk}$ of the $k$-tied Normal distribution so that after the outer-product operation the respective standard deviations $\sigma_{ij}$ have the same mean values as we obtain in the standard GMFVI posterior parametrization. In the experiments for this section we use KL annealing where we linearly scale-up the contribution of the KL term from a fraction of its full value to its full contribution over the course of training. Finally, similarly to the previous section, we picked the key hyper-parameters such as the KL annealing rate, a learning rate, a batch size and a prior standard deviation based on the performance on respective validation sets. For more information about the experimental details for this section please refer to Appendix A.2.

## 3.2 EXPERIMENTAL RESULTS

We first investigate the comparison of predictive performance of GMFVI Bayesian neural network models trained using the $k$-tied Normal posterior distribution with the models trained using the standard parametrization of the GMFVI. Figure 4 shows the results for the three model types on the test splits of their respective datasets. We analyse variants of the $k$-tied Normal posterior distribution with different levels of tying $k$. We observe that for $k \geq 2$ the $k$-tied Normal posterior is able to achieve the performance competitive with the standard GMFVI posterior parametrization, while reducing the total number of model parameters. The benefits of using the $k$-tied Normal posterior are the most visible for models where the dense layers with the $k$-tied Normal posterior constitute for a significant portion of the total number of the model parameters (e.g. MLPs and CNNs with dense layers for classification).

We further investigate the impact of the $k$-tied Normal posterior distribution on the signal-to-noise ratio[2] (SNR) of stochastic gradient estimates of the variational lower bound (ELBO). In particular, we focus on the gradient SNR of the GMFVI posterior standard deviation parameters for which we perform the tying. These parameters are either $u_{ik}$ or $v_{jk}$ for the $k$-tied Normal posterior or $\sigma_{ij}$ for the standard GMFVI parametrization, all optimized in their log forms for numerical stability. The SNR results in the Figure 4 show a significant increase in the gradient SNR when using the $k$-tied Normal posterior.

Consequently, we observe that the increase in the gradient SNR translates into faster convergence of the negative ELBO objective in some of the analyzed models. Figure 5 shows the convergence plots of validation negative ELBO for all the three model types. We observe that the impact of the $k$-tied Normal posterior on convergence depends on the model type. As shown in Figure 4, the impact on the MLP model is strong and consistent with the $k$-tied Normal posterior increasing convergence speed compared to the standard GMFVI parameterization. For the LSTM model we also observe a similar speed-up. However, for the CNN model the impact of the $k$-Normal posterior on the ELBO convergence is much smaller. We hypothesize that this is due to the fact that we use the $k$-tied Normal posterior for all the layers trained using GMFVI in the MLP and the LSTM models, while in the CNN model we use the $k$-tied Normal posterior only for some of the GMFVI

---

[2]SNR for each gradient value is calculated as $E[g_b^2]/\mathrm{Var}[g_b^2]$, where $g_b$ is the gradient value for a single parameter. The expectation $E$ and variance $Var$ of the gradient values $g_b$ are calculated over a window of last 10 batches.

trained layers. More precisely, in the CNN model we use the $k$-tied Normal posterior only for the two dense layers, while the two convolutional layers are trained using the standard parameterization of the GMFVI. Finally, we point out that the $k$-tied Normal posterior does not increase the training step time compared to the standard parameterization of the GMFVI. See Table 4 in Appendix D for the support of this claim.

| Model & Dataset | Rank $k$ | -ELBO ↓ | NLL ↓ | Accuracy ↑ | #Par. [k] ↓ |
|---|---|---|---|---|---|
| MNIST, MLP | full | $0.501_{\pm 0.0061}$ | $0.133_{\pm 0.0040}$ | $96.8_{\pm 0.18}$ | 957 |
| MNIST, MLP | 1 | $0.539_{\pm 0.0063}$ | $0.155_{\pm 0.0043}$ | $96.1_{\pm 0.19}$ | 482 |
| MNIST, MLP | 2 | $0.520_{\pm 0.0063}$ | $0.129_{\pm 0.0039}$ | $96.8_{\pm 0.18}$ | 484 |
| MNIST, MLP | 3 | $0.497_{\pm 0.0060}$ | $0.120_{\pm 0.0038}$ | $96.9_{\pm 0.18}$ | 486 |
| CIFAR100, CNN | full | $3.72_{\pm 0.018}$ | $2.16_{\pm 0.016}$ | $43.9_{\pm 0.50}$ | 4,405 |
| CIFAR100, CNN | 1 | $3.65_{\pm 0.017}$ | $2.12_{\pm 0.015}$ | $45.5_{\pm 0.50}$ | 2,262 |
| CIFAR100, CNN | 2 | $3.76_{\pm 0.019}$ | $2.15_{\pm 0.016}$ | $44.3_{\pm 0.50}$ | 2,268 |
| CIFAR100, CNN | 3 | $3.73_{\pm 0.018}$ | $2.13_{\pm 0.016}$ | $44.3_{\pm 0.50}$ | 2,273 |
| IMDB, LSTM | full | $0.538_{\pm 0.0054}$ | $0.478_{\pm 0.0052}$ | $79.5_{\pm 0.26}$ | 2,823 |
| IMDB, LSTM | 1 | $0.592_{\pm 0.0041}$ | $0.512_{\pm 0.0040}$ | $77.6_{\pm 0.26}$ | 2,693 |
| IMDB, LSTM | 2 | $0.560_{\pm 0.0042}$ | $0.484_{\pm 0.0041}$ | $78.2_{\pm 0.26}$ | 2,694 |
| IMDB, LSTM | 3 | $0.550_{\pm 0.0051}$ | $0.491_{\pm 0.0050}$ | $78.8_{\pm 0.26}$ | 2,695 |

| Rank $k$ | MNIST, MLP Dense 2, SNR at step | | |
|---|---|---|---|
| | 1000 | 5000 | 9000 |
| full | $4.13_{\pm 0.027}$ | $4.45_{\pm 0.091}$ | $3.21_{\pm 0.035}$ |
| 1 | $5840_{\pm 190}$ | $158_{\pm 3.8}$ | $5.3_{\pm 0.20}$ |
| 2 | $7500_{\pm 240}$ | $140_{\pm 11}$ | $4.3_{\pm 0.26}$ |
| 3 | $7000_{\pm 270}$ | $117_{\pm 1.7}$ | $4.1_{\pm 0.20}$ |

| Rank $k$ | MNIST, MLP, -ELBO at step | | |
|---|---|---|---|
| | 1000 | 5000 | 9000 |
| full | $42.16_{\pm 0.070}$ | $26.52_{\pm 0.016}$ | $15.39_{\pm 0.016}$ |
| 1 | $43.11_{\pm 0.039}$ | $14.85_{\pm 0.017}$ | $2.06_{\pm 0.027}$ |
| 2 | $42.74_{\pm 0.090}$ | $13.97_{\pm 0.023}$ | $1.82_{\pm 0.017}$ |
| 3 | $42.63_{\pm 0.068}$ | $13.61_{\pm 0.020}$ | $1.80_{\pm 0.031}$ |

**Figure 4:** Left: impact of the $k$-tied Normal posterior on test ELBO, test predictive performance and number of model parameters. Test performance is reported as a mean and SEM across 100 weights samples after training each model for $\approx$300 epochs. Right top: mean gradient SNR in the Dense 2 layer of the MNIST MLP model at increasing training steps for different ranks of tying $k$. We observe a similar increase in the SNR from tying for the CNN and the LSTM models as for the MLP model shown here. We report mean and SEM across 3 training runs with different random seeds. Right bottom: Negative ELBO on the MNIST validation data set at increasing training steps for different ranks of tying $k$. See also Figure 5, which shows negative ELBO convergence plots for the all three models types.

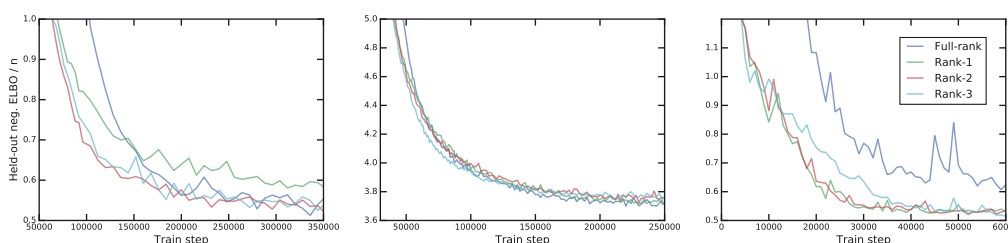

**Figure 5:** Convergence of negative ELBO (lower is better) reported for validation dataset when training with tied variational posterior standard deviations for MLP (left), CNN (center), and LSTM (right) with different low-rank factorizations of the posterior standard deviation matrix. Full-rank is the standard parametrization of the GMFVI.

# 4 RELATED WORK

The application of variational inference to neural networks dates back at least to Peterson (1987); Hinton & Van Camp (1993). Many developments[3] have followed those seminal research efforts, in particular regarding (1) the expressiveness of the variational posterior distribution and (2) the way the variational parameters themselves can be structured to lead to compact, easier-to-learn and scalable formulations. We organize the discussion of this section around those two aspects, with a specific focus on the Gaussian case.

**Full Gaussian posterior.** Because of their substantial memory and computational cost, Gaussian variational distributions with full covariance matrices have been primarily applied to (generalized) linear models and shallow neural networks (Jaakkola & Jordan, 1997; Barber & Bishop, 1998; Marlin et al., 2011; Titsias & Lázaro-Gredilla, 2014; Miller et al., 2017; Ong et al., 2018).

To represent the dense covariance matrix efficiently in terms of variational parameters, several schemes have been proposed, including the sum of low-rank plus diagonal matrices (Barber & Bishop, 1998; Seeger, 2000; Miller et al., 2017; Zhang et al., 2017; Ong et al., 2018), the Cholesky decomposition (Challis & Barber, 2011) or by operating instead on the precision matrix (Tan & Nott, 2018; Mishkin et al., 2018).

---

[3]We refer the interested readers to Zhang et al. (2018) for a recent review of variational inference.

**Gaussian posterior with block-structured covariances.** In the context of Bayesian neural networks, the layers represent a natural structure to be exploited by the covariance matrix. When assuming independence across layers, the resulting covariance matrix exhibits a *block-diagonal structure* that has been shown to be a well-performing simplification of the dense setting (Sun et al., 2017; Zhang et al., 2017), with both memory and computational benefits.

Within each layer, the corresponding diagonal block of the covariance matrix can be represented by a Kronecker product of two smaller matrices (Louizos & Welling, 2016; Sun et al., 2017), possibly with a parametrization based on rotation matrices (Sun et al., 2017). Finally, using similar techniques, Zhang et al. (2017) proposed to use a block tridiagonal structure that better approximates the behavior of a dense covariance.

**Fully factorized mean-field Gaussian posterior.** A fully factorized Gaussian variational distribution constitutes the simplest option for variational inference. The resulting covariance matrix is diagonal and all underlying parameters are assumed to be independent. While the mean-field assumption is known to have some limitations—e.g., underestimated variance of the posterior distribution (Turner & Sahani, 2011) and robustness issues (Giordano et al., 2018)—it leads to scalable formulations, with already competitive performance, as for instance illustrated by the recent uncertainty quantification benchmark of Ovadia et al. (2019).

Because of its simplicity and scalability, the fully-factorized Gaussian variational distribution has been widely used for Bayesian neural networks (Graves, 2011; Ranganath et al., 2014; Blundell et al., 2015; Hernández-Lobato & Adams, 2015; Zhang et al., 2017; Khan et al., 2018).

Our approach can be seen as an attempt to further reduce the number of parameters of the (already) diagonal covariance matrix. Closest to our approach is the work of Louizos & Welling (2016). Their matrix variate Gaussian distribution instantiated with the Kronecker product of the diagonal row- and column-covariance matrices leads to a rank-1 tying of the posterior variances. In contrast, we explore tying strategies beyond the rank-1 case, which we show to lead to better performance (both in terms of ELBO and predictive metrics). Importantly, we further prove that tying strategies with a rank greater than one cannot be represented in a matrix variate Gaussian distribution, thus clearly departing from (Louizos & Welling, 2016) (see Appendix B for details).

Our approach can be also interpreted as a particular case of *hierarchical* variational inference (Ranganath et al., 2016) where the prior on the variational parameters corresponds to a Dirac distribution, non-zero only when a pre-specified low-rank tying relationship holds.

We close this related work section by mentioning the existence of other strategies to produce more flexible approximate posteriors, e.g., normalizing flows (Rezende & Mohamed, 2015) and extensions thereof (Louizos & Welling, 2017).

## 5 CONCLUSION

In this work we have shown that Bayesian Neural Networks trained with standard Gaussian Mean-Field Variational Inference learn posterior standard deviation matrices that can be approximated with little information loss by low-rank SVD decompositions. This suggests that richer parameterizations of the variational posterior may not always be better, and that compact parameterizations can also work well. We used this insight to propose a simple, yet effective variational posterior parametrization, which speeds up training and reduces the number of variational parameters without degrading predictive performance on 3 different model types.

In future work, we hope to scale up variational inference with compactly parameterized approximate posteriors to much larger models and more complex problems. For mean-field variational inference to work well in that setting several challenges will likely need to be addressed (Osawa et al., 2019); improving the signal-to-noise ratio of ELBO gradients using our compact variational parameterizations may provide a piece of the puzzle.

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

# A  ADDITIONAL DETAILS

## A.1  EXPERIMENTAL DETAILS FOR SECTION 2

We provide here some of the experimental details that we skipped over in the main text of the Section 2 of the paper.

More architecture details:

- The 3 layers of the MLP have number of units respectivelly 400, 400 and 10.
- The 2 convolutional layers of the CNN have filters of sizes 32 and 64 and 2 dense layers with the number of units 512 and 100.
- The embedding and LSTM cell in the LSTM model are both of size 128.
- In the MLPs and the CNN models we train both the kernel and bias weights using GMFVI. In the LSTM model we train only the kernel weights using GMFVI and we trained the bias weights using MAP estimation.
- It is possible that the SVD decomposition of posterior standard deviation matrix will result in a low-rank approximation that contains negative values. In such cases, we threshold the minimum values of the resulting approximation at very low positive constant.

Details of the GMFVI optimisation:

- In the MLPs and the CNN models we train both the kernel and bias weights using GMFVI. In the LSTM model we train only the kernel weights using GMFVI and we trained the bias weights using MAP estimation.
- For each of the three models the GMFVI uses the standard reparametrization trick (Kingma & Welling, 2013).
- It is possible that the SVD decomposition of posterior standard deviation matrix will result in a low-rank approximation that contains negative values. In such cases, we threshold the minimum values of the resulting approximation at very low positive constant.
- We initialize the variational posterior means using the standard He initialization (He et al., 2015) and the posterior standard deviations using samples from $\mathcal{N}(0.01, 0.001)$.
- We use a Normal prior $\mathcal{N}(0, \sigma_p)$ for all the weights and select the $\sigma_p$ for each of the models separately from a set of $[0.2, 0.3]$ based on validation data perforfmance.
- For optimization we used an Adam optimizer (Kingma & Ba (2014)), for which we picked the optimal learning rate for each model from the set of $[0.0001, 0.0003, 0.001, 0.003]$ based on validation set performance.

Additional details:

- It is possible that the SVD decomposition of posterior standard deviation matrix will result in a low-rank approximation that contains negative values. In such cases, we threshold the minimum values of the resulting approximation at very low positive constant.

## A.2 EXPERIMENTAL DETAILS FOR SECTION 3

We provide here some of the experimental details that we skipped over in the main text of the Section 3 of the paper:

- For the $k$-tied Normal posteriors to work reliably we used KL annealing with a linearly increasing scaling of the KL term during training. We select the best linear coefficient of the KL annealing from $5 \times 10^{-5}$ and $5 \times 10^{-6}$ per batch and update it's value every 100 batches.
- We initalize the parameters $u_{ik}$ and $v_{jk}$ so that after the outer-product operation the respective $\sigma_{ij}$ standard deviations have means at $0.01$ before transforming to log-domain. That means that in the log domain the parameters $u_{ik}$ and $v_{jk}$ are initialized as $0.5(\log(0.01) - \log(k))$. We also add white noise $\mathcal{N}(0, 0.1)$ to the values of $u_{ik}$ and $v_{jk}$ in the log domain to break symmetry.

## B PROOF OF THE MATRIX VARIATE NORMAL PARAMETERIZATION

In this section of the appendix, we formally explain the connections between the $k$-tied Normal distribution and the matrix variate Gaussian distribution (Gupta & Nagar, 2018), referred to as $\mathcal{MN}$.

Consider positive definite matrices $\mathbf{Q} \in \mathbb{R}^{r \times r}$ and $\mathbf{P} \in \mathbb{R}^{c \times c}$ and some arbitrary matrix $\mathbf{M} \in \mathbb{R}^{r \times c}$. We have by definition that $\mathbf{W} \in \mathbb{R}^{r \times c} \sim \mathcal{MN}(\mathbf{M}, \mathbf{Q}, \mathbf{P})$ if and only if $\text{vec}(\mathbf{W}) \sim \mathcal{N}(\text{vec}(\mathbf{M}), \mathbf{P} \otimes \mathbf{Q})$, where $\text{vec}(\cdot)$ stacks the columns of a matrix and $\otimes$ is the Kronecker product

The $\mathcal{MN}$ has already been used for variational inference by Louizos & Welling (2016) and Sun et al. (2017). In particular, Louizos & Welling (2016) consider the case where both $\mathbf{P}$ and $\mathbf{Q}$ are restricted to be diagonal matrices. In that case, the resulting distribution corresponds to our $k$-tied Normal distribution with $k = 1$ since

$$\mathbf{P} \otimes \mathbf{Q} = \text{diag}(\mathbf{p}) \otimes \text{diag}(\mathbf{q}) = \text{diag}(\text{vec}(\mathbf{q}\mathbf{p}^\top)).$$

Importantly, we prove below that, in the case where $k \geq 2$, the $k$-tied Normal distribution cannot be represented as a matrix variate Gauussian distribution.

**Lemma B.1** (Rank-2 matrix and Kronecker product). *Let $\mathbf{B}$ be a rank-2 matrix in $\mathbb{R}_+^{r \times c}$. There do not exist matrices $\mathbf{Q} \in \mathbb{R}^{r \times r}$ and $\mathbf{P} \in \mathbb{R}^{c \times c}$ such that*

$$diag(vec(\mathbf{B})) = \mathbf{P} \otimes \mathbf{Q}.$$

*Proof.* Let us introduce the shorthand $\mathbf{D} = \text{diag}(\text{vec}(\mathbf{B}))$. By construction, $\mathbf{D}$ is diagonal and has its diagonal terms strictly positive (it is assumed that $\mathbf{B} \in \mathbb{R}_+^{r \times c}$, i.e., $b_{ij} > 0$ for all $i, j$).

We proceed by contradiction. Assume there exist $\mathbf{Q} \in \mathbb{R}^{r \times r}$ and $\mathbf{P} \in \mathbb{R}^{c \times c}$ such that $\mathbf{D} = \mathbf{P} \otimes \mathbf{Q}$.

This implies that all diagonal blocks of $\mathbf{P} \otimes \mathbf{Q}$ are themselves diagonal with strictly positive diagonal terms. Thus, $p_{jj}\mathbf{Q}$ is diagonal for all $j \in \{1, \ldots, c\}$, which implies in turn that $\mathbf{Q}$ is diagonal, with non-zero diagonal terms and $p_{jj} \neq 0$. Moreover, since the off-diagonal blocks $p_{ij}\mathbf{Q}$ for $i \neq j$ must be zero and $\mathbf{Q} \neq \mathbf{0}$, we have $p_{ij} = 0$ and $\mathbf{P}$ is also diagonal.

To summarize, if there exist $\mathbf{Q} \in \mathbb{R}^{r \times r}$ and $\mathbf{P} \in \mathbb{R}^{c \times c}$ such that $\mathbf{D} = \mathbf{P} \otimes \mathbf{Q}$, then it holds that $\mathbf{D} = \text{diag}(\mathbf{p}) \otimes \text{diag}(\mathbf{q})$ with $\mathbf{p} \in \mathbb{R}^c$ and $\mathbf{q} \in \mathbb{R}^r$. This last equality can be rewritten as $b_{ij} = p_j q_i$ for all $i \in \{1, \ldots, r\}$ and $j \in \{1, \ldots, c\}$, or equivalently

$$\mathbf{B} = \mathbf{q}\mathbf{p}^\top.$$

This leads to a contradiction since $\mathbf{q}\mathbf{p}^\top$ has rank one while $\mathbf{B}$ is assumed to have rank two. □

Figure 6 provides an illustration of the difference between the $k$-tied Normal and the $\mathcal{MN}$ distribution.

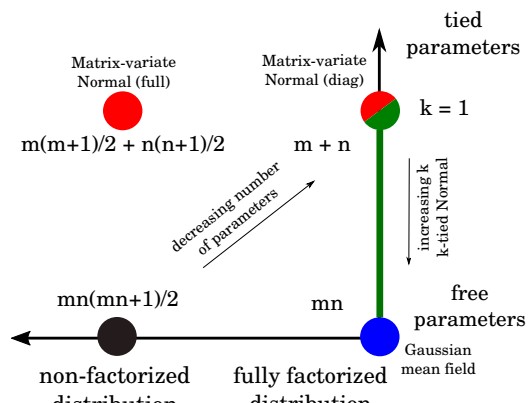

**Figure 6:** Illustration of the difference between the $k$-tied Normal (green), the $\mathcal{MN}$ distribution (red), the Gaussian mean field (blue) and the full Gaussian covariance (black) for a matrix of posterior standard deviations for a layer of size $m \times n$. $k$-tied Normal with $k = 1$ is equivalent to $\mathcal{MN}$ with diagonal row and column covariance matrices (half-red, half-green circle). Our experiments show that the $k = 1$ fails to capture the performance of the mean field. On the other hand, while the full/non-diagonal $\mathcal{MN}$ increases the expressiveness of the posterior, it also increases the number of parameters. In contrast, $k$-tied Normal with $k \geq 2$ not only decreases the number of parameters, but also matches the predictive performance of the mean field.

## C  LOW-RANK STRUCTURE IN POSTERIOR STANDARD DEVIATIONS OF CONVOLUTIONAL LAYERS

Encouraged by the reviewers, we analyze the applicability of our method to different layer types and larger models. In particular, we investigate the low-rank structure of the posterior standard deviations of both the dense layers and the convolutional layers in a ResNet-18 (He et al., 2016) GMFVI Bayesian neural network trained on the CIFAR-10 dataset (Krizhevsky et al., 2009a). More precisely, we use a publicly available model[4] from the example repository of Tensorflow Probability (Dillon et al., 2017). To perform the low-rank decomposition of the convolutional layers, we flatten all the dimensions of the convolutional layers except the last dimension (e.g., a layer with shape $[3, 3, 512, 512]$ is reshaped to $[3 \cdot 3 \cdot 512, 512]$). See Figure 7 for the examples of the resulting 2-dimensional matrices. We then analyze the SVD spectrum of the posterior means and standard deviations of such flattened convolutional layers. Interestingly, we observe a similar low-rank structure in the flattened convolutional layers as observed in the dense layers, see Figure 8. The low-rank structure is the most visible for the last convolutional layer, which also contain the highest number of parameters.

Importantly, note that after performing the low-rank approximation in the 2-dimensional space, we can reshape the materialized 2-dimensional matrices back into the 4-dimensional form of a convolutional layer. Table 3 shows that such a low-rank approximation in the convolutional layers of the analyzed ResNet-18 model can be performed without a loss in the model's predictive performance, while halving the total number of parameters. These results suggest that our method can be applicable not only to the different model types (MLP, CNN and LSTM), but also to a range of layer types (dense and convolutional) in large and practically applicable models such as the ResNet-18 model.

| Rank | -ELBO ↓ | NLL ↓ | Accuracy ↑ | #Params ↓ | %Params ↓ |
|------|---------|-------|-----------|-----------|-----------|
| Full | $122.61_{\pm 0.012}$ | $0.495_{\pm 0.0080}$ | $83.5_{\pm 0.37}$ | 9,814,026 | 100.0 |
| 1 | $122.57_{\pm 0.012}$ | $0.658_{\pm 0.0069}$ | $81.7_{\pm 0.39}$ | 4,929,711 | 50.2 |
| 2 | $122.77_{\pm 0.012}$ | $0.503_{\pm 0.0080}$ | $83.2_{\pm 0.37}$ | 4,946,964 | 50.4 |
| 3 | $122.67_{\pm 0.012}$ | $0.501_{\pm 0.0079}$ | $83.2_{\pm 0.37}$ | 4,964,217 | 50.6 |

**Table 3:** Impact of the low-rank approximation of the GMFVI-trained posterior standard deviations of a ResNet-18 model on the model's ELBO and predictive performance. We report mean and SEM of each metric across 100 models samples.

---

[4]`https://github.com/tensorflow/probability/blob/master/tensorflow_`
`probability/examples/cifar10_bnn.py`

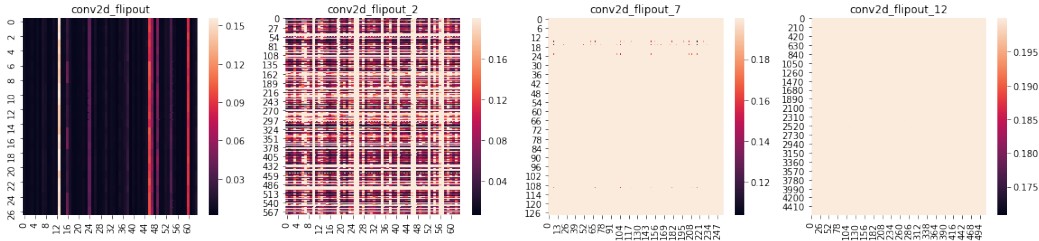

**Figure 7:** Heat maps of the partially flattened posterior standard deviation tensors for the selected convolutional layers of the ResNet-18 GMFVI BNN trained on CIFAR-10. The partially flattened posterior standard deviation tensors of the convolutional layers display similar low-rank patterns that we observe for the dense layers.

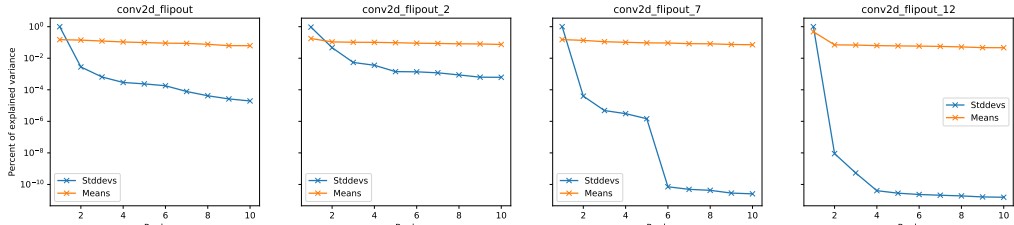

**Figure 8:** Explained variance per singular value from SVD of partially flattened tensors of posterior means and posterior standard deviations for different convolutional layers of the ResNet-18 GMFVI BNN trained on CIFAR-10. Posterior standard deviations clearly display strong low-rank structure, with most of the variance contained in the top few singular values, while this is not the case for posterior means.

# D   ADDITIONAL EXPERIMENTAL RESULTS

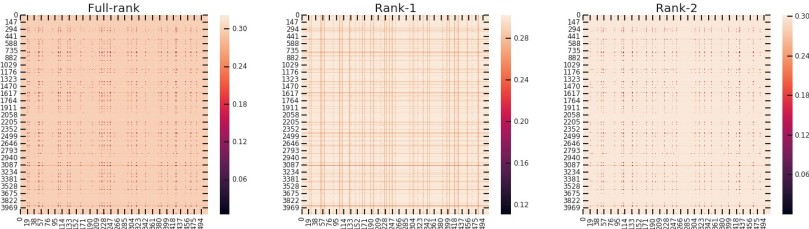

**Figure 9:** Heat map of the posterior standard deviation matrix for the weights in the first dense layer of a LeNet CNN trained using GMFVI on the CIFAR-100 dataset (left), as well as its rank-1 approximation (middle) and rank-2 approximation (right). The rank-2 approximation looks visually similar to the full-rank matrix, confirming our numerical results from Figure 2.

| Training method | Train step time [ms] $\downarrow$ |
|---|---|
| Point estimate | $2.00_{\pm 0.0064}$ |
| Standard GMFVI | $7.17_{\pm 0.014}$ |
| K-tied GMFVI | $6.14_{\pm 0.018}$ |

**Table 4:** Training step evaluation times for a simple model architecture with two dense layers[5] for different training methods. We report mean and SEM across a single training run. The $k$-tied Normal posterior does not increase the train step evaluation times compared to the standard parameterization of the GMFVI posterior. We expect this to hold more generally because the biggest additional operation per step when using the $k$-tied Normal posterior is the $\mathbf{U}\mathbf{V}^T$ multiplication to materialize the matrix of posterior standard deviations $\mathbf{A}$, where $\mathbf{U} \in \mathbb{R}^{m \times k}$, $\mathbf{V} \in \mathbb{R}^{n \times k}$ and $k$ is a small value (e.g., 2 or 3). The time complexity of this operations is $\mathcal{O}(kmn)$, which is usually negligible compared to the time complexity of data-weight matrix multiplication $\mathcal{O}(bmn)$, where $b$ is the batch size.

---

[5]E.g., as used in `https://github.com/tensorflow/docs/blob/master/site/en/tutorials/keras/classification.ipynb`. Our GMFVI implementation of this model is available under `https://colab.research.google.com/drive/14pqe_VG5s49xlcXB-Jf8S9GoTFyjv4OF`

