# OpenReview forum: "On the Parameterization of Gaussian Mean Field Posteriors in Bayesian Neural Networks"
_ICLR.cc/2020/Conference — Reject_

### Official Review · AnonReviewer3 · 2019-10-20
**Official Blind Review #3**

**Rating:** 3

**Review:**

This paper showed the (diagonal) variance parameters in mean-field VI for BNNs exhibit a low-rank structure, and that training from scratch using such a low-rank parameterization lead to comparable performance as well as increased SNR of the gradient.

While the observation is somewhat interesting, currently it is only verified in a narrow range of network architectures, and it's unclear if the observation and the proposed method will still be useful on network architectures used in real-world applications. As such, I believe this work would be more suitable as a workshop presentation.

More specifically, the models considered are MLP on MNIST, LeNet on CIFAR-100 and LSTM on IMDB. These choices are not practical, as the reported performance indicates (e.g. 45% accuracy on CIFAR-100); as such these results cannot support the claim that the proposed low-rank parameterization could be useful in practice: while MFVI can be useful on some model architectures, it lead to pathologies on others, especially on smaller networks. See Fig.1 in [1] for an example and [2] for a possible explanation. Also note that the reported accuracy on MNIST is ~2% worse than the typical values in BNN papers using a comparable setting, e.g. [3]. These facts unfortunately lead to the doubt that the proposed low-rank parameterization could only match the performance of MFVI when MFVI is not that useful.

Another major concern is that I'm not sure if the proposed low-rank variational would actually save parameters in practice, since the variance parameter in MFVI could already be stored as the preconditioners in Adam-like optimizers [4-5].

Suggestions for future improvement:
* Re-do the experiments using more complex network architectures, and optionally, on larger datasets / more complex tasks (e.g. image segmentation as in [6]).
* Also, consider setups more commonly used in previous BNN papers, e.g. VGG/ResNet on CIFAR-10 has been used in [4,7,8]. CIFAR-100 could be sufficiently complex as a BNN benchmark, but few papers reported results on it.
* Report the quality of the learned uncertainty, either directly as in [9] or using performance of downstream tasks, e.g. RL and adversarial robustness as in [4,8].

# References
[1] Structured and Efficient Variational Deep Learning with Matrix Gaussian Posteriors
[2] Overpruning in Variational Bayesian Neural Networks
[3] A Unified Particle-Optimization Framework for Scalable Bayesian Sampling
[4] Noisy Natural Gradient as Variational Inference
[5] Fast and Scalable Bayesian Deep Learning by Weight-Perturbation in Adam
[6] Bayesian Uncertainty Estimation for Batch Normalized Deep Networks
[7] Learning Weight Uncertainty with Stochastic Gradient MCMC for Shape Classification
[8] Function Space Particle Optimization for Bayesian Neural Networks
[9] Can You Trust Your Model’s Uncertainty? Evaluating Predictive Uncertainty Under Dataset Shift

**Experience Assessment:**

I have published one or two papers in this area.

**Review Assessment: Checking Correctness Of Derivations And Theory:**

I assessed the sensibility of the derivations and theory.

**Review Assessment: Checking Correctness Of Experiments:**

I carefully checked the experiments.

**Review Assessment: Thoroughness In Paper Reading:**

I read the paper thoroughly.

---

> ### Author Response · Authors · 2019-11-15
> **Response to AnonReviewer3**
>
> [R3.1]
> While the observation is somewhat interesting, currently it is only verified in a narrow range of network architectures, and it's unclear if the observation and the proposed method will still be useful on network architectures used in real-world applications. As such, I believe this work would be more suitable as a workshop presentation.
>
> More specifically, the models considered are MLP on MNIST, LeNet on CIFAR-100 and LSTM on IMDB. These choices are not practical, as the reported performance indicates (e.g. 45% accuracy on CIFAR-100); as such these results cannot support the claim that the proposed low-rank parameterization could be useful in practice: while MFVI can be useful on some model architectures, it lead to pathologies on others, especially on smaller networks. See Fig.1 in [1] for an example and [2] for a possible explanation. Also note that the reported accuracy on MNIST is ~2% worse than the typical values in BNN papers using a comparable setting, e.g. [3]. These facts unfortunately lead to the doubt that the proposed low-rank parameterization could only match the performance of MFVI when MFVI is not that useful.
>
> [R3.1 our response]
> Encouraged by the reviewer, we investigated the behaviour on a larger ResNet-18 model (provided by https://github.com/tensorflow/probability/blob/master/tensorflow_probability/examples/cifar10_bnn.py) and demonstrate that the approximation quality claimed in our paper holds also for convolution layers and for the deeper ResNet-18 model (see Figure 3 in Section 2 of the updated paper). We are therefore confident the finding is of broad applicability and plan to include further experiments in the final version of the paper.
>
> The 2% gap in the MNIST accuracies is due to the difference in the training procedure we used and the procedures which are commonly used in other MFVI papers. We train the models until full ELBO convergence, without early stopping (which is commonly used in the MFVI literature). Early stopping can increase the validation accuracy by e.g. 2% compared to the accuracy at full convergence. The reason for this is that the MFVI models tend to start under-fitting as training progresses when using the full contribution of the KL term. This is due to the fact that the posterior variances increase to reach the prior variance and introduce large amount of noise which reduces how well the model can fit the training data. This is a limitation of the MFVI approach more generally and not specific to our method, but we agree that it is important to understand and address this shortcoming of the MFVI approach in future research.
>
> [R3.2]
> Another major concern is that I'm not sure if the proposed low-rank variational would actually save parameters in practice, since the variance parameter in MFVI could already be stored as the preconditioners in Adam-like optimizers [4-5].
>
> [R3.2 our response]
> We agree that if preconditioning is feasible then the training time memory savings are irrelevant.  However, after training the preconditioner is typically discarded and for test-time inference our method does approximately halve the required model size as shown in Tables 2 and 5.
>
> [R3.3]
> Suggestions for future improvement:
> * Re-do the experiments using more complex network architectures, and optionally, on larger datasets / more complex tasks (e.g. image segmentation as in [6]).
> * Also, consider setups more commonly used in previous BNN papers, e.g. VGG/ResNet on CIFAR-10 has been used in [4,7,8]. CIFAR-100 could be sufficiently complex as a BNN benchmark, but few papers reported results on it.
> * Report the quality of the learned uncertainty, either directly as in [9] or using performance of downstream tasks, e.g. RL and adversarial robustness as in [4,8].
>
> [R3.3 our response]
> As the reviewer suggested, we extended the experiments using a more complex network architecture (ResNet) trained on a commonly used CIFAR-10 benchmark. In the future, we also plan to extend the analysis to report the quality of the learned uncertainty as the reviewer suggested.

---

### Official Review · AnonReviewer2 · 2019-10-22
**Official Blind Review #2**

**Rating:** 1

**Review:**

The paper considers variational Bayesian inference (learning) for neural networks assuming that both the prior distributions and the posterior distributions of the network weights are factorising Gaussians. It is well known that the respective optimisation task for the parameters of the posterior weight distributions (ELBO) is tractable by stochastic gradient ascent.  The authors propose to simplify (restrict) the model even further, by assuming that the posterior variances for fully connected layers, (when seen as a matrix) have low rank. It is pretty obvious that the ELBO optimisation task remains tractable in this case.

The authors perform two types of experiments to show that the proposed simpler model does not decrease the performance of the network, when compared to the full rank factorising model. They first show that the learned variances of dense layers indeed exhibit a low rank structure for three models learned on the corresponding data (MLP on MNIST, LeNet on CIFAR and LSTM on IMDB). Then, in a second step, they consider the proposed rank constraint during learning and show that with rank k >= 2 the models are able to achieve performance competitive with the full rank model and, moreover, exhibit better signal to noise ratio for the gradients during learning.

The paper is well written, clearly structured and technically correct. However, in my opinion, its novelty and new insights are restricted, which is why I suggest to reject the paper. The reasons for this are the following.

- The authors restrict their analysis to dense layers only. Moreover, it remains conceptually unclear, why and when the proposed model should be useful and represent a good approximation of the full rank model.

- The experimental study is restricted to small models, e.g. in the case of CIFAR a quite shallow version of LeNet which achieves only ~45% validation accuracy. It remains unclear, whether the observed low rank structure of the variance matrices (of the dense layers) will scale to deeper models that could achieve competitive validation accuracies.

- When measuring the impact of the low rank tying of the posterior variances, the authors compare to the full rank model only. I am missing the "point estimate" baseline for these models. For if the the positive impact of the Bayesian inference approach with the full rank model is small when compared to the point estimate, then, as a consequence, the impact of the low rank tying must be small when compared to the full rank model.

- The numbers reported in Table 3 (second group of experiments) raise some questions not addressed by the authors. It remains unclear, why the low rank model with ranks 1,2,3 gives better accuracies on CIFAR than the full rank model. The same holds for the LSTM experiment. This may indicate some kind of "overfitting" and should have been analysed by the authors in order to "disentangle" possible overfitting issues from the question of validity of the proposed low rank model.

**Experience Assessment:**

I have published one or two papers in this area.

**Review Assessment: Checking Correctness Of Derivations And Theory:**

I carefully checked the derivations and theory.

**Review Assessment: Checking Correctness Of Experiments:**

I carefully checked the experiments.

**Review Assessment: Thoroughness In Paper Reading:**

I read the paper thoroughly.

---

> ### Author Response · Authors · 2019-11-15
> **Response to AnonReviewer2**
>
> [R2.1]
> - The authors restrict their analysis to dense layers only. Moreover, it remains conceptually unclear, why and when the proposed model should be useful and represent a good approximation of the full rank model.
>
> - The experimental study is restricted to small models, e.g. in the case of CIFAR a quite shallow version of LeNet which achieves only ~45% validation accuracy. It remains unclear, whether the observed low rank structure of the variance matrices (of the dense layers) will scale to deeper models that could achieve competitive validation accuracies.
>
> [R2.1 our response]
> Encouraged by the reviewer, we investigated the behaviour on a larger ResNet-18 model (provided by https://github.com/tensorflow/probability/blob/master/tensorflow_probability/examples/cifar10_bnn.py) and demonstrate that the approximation quality claimed in our paper for the dense layers of the MLP, CNN and LSTM models holds also for the convolution layers and of the deeper ResNet-18 model (see Figure 3 in Section 2 and Appendix C of the updated paper). We are therefore confident the finding is of broad applicability and plan to include further experiments in the final version of the paper.
>
> Exploring the theoretical justification for the observed phenomena remains an interesting future work. In particular, we are currently considering simpler settings, e.g., shallow Bayesian neural networks with linear activation functions, where some derivations can be obtained in closed forms, which constitutes a good starting point for a theoretical analysis.
>
> [R2.2]
> When measuring the impact of the low rank tying of the posterior variances, the authors compare to the full rank model only. I am missing the "point estimate" baseline for these models. For if the the positive impact of the Bayesian inference approach with the full rank model is small when compared to the point estimate, then, as a consequence, the impact of the low rank tying must be small when compared to the full rank model.
>
> [R2.2 our response]
> The goal of our paper is not to prove the positive impact of the Bayesian inference or the low rank tying over the point estimate models. In contrast, we start from the observation that scaling up mean-field and making it more robust is still a challenging problem ([0]). We try to tackle this problem by reducing the number of parameters to optimize---while maintaining a comparable predictive behavior---and obtaining less noisy gradient estimates.
>
> [R2.3]
> The numbers reported in Table 3 (second group of experiments) raise some questions not addressed by the authors. It remains unclear, why the low rank model with ranks 1,2,3 gives better accuracies on CIFAR than the full rank model. The same holds for the LSTM experiment. This may indicate some kind of "overfitting" and should have been analysed by the authors in order to "disentangle" possible overfitting issues from the question of validity of the proposed low rank model.
>
> [R2.3 our response]
> We are confident that the results are not influenced by the "overfitting" issues. By varying k of the k-tied Normal, we are varying the parameterization of the posterior distribution, not the original model. Restricting the posterior approximation will not lessen overfitting. In fact, the point estimate (zero variance) will overfit most of all posterior approximations. We hypothesise that the improvement in the CNN model from using the low-rank model can be attributed to the improved learning dynamics (higher gradient signal-to-noise ratio).

---

### Official Review · AnonReviewer1 · 2019-10-24
**Official Blind Review #1**

**Rating:** 3

**Review:**

The paper proposes a low-rank approximation to the diagonal of Gaussian mean field posterior which reduces the number of parameters to fit. They show that the predictive performance doesn't drop much compared with the full covariance but the number of parameters is significantly reduced.

1. Why Matrix normal distribution is related to k-tied Normal distribution when k=1. When k=1, the rank of UV^\top is 1. The covariance is matrix normal is U\otimes V, whose rank is rank(U)rank(V). Also MN has a full covariance for the Gaussian approximation, not mean field. MN is only equal to 1-tied when only diagonal row and column covariances are considered. If that's what the paper means, k-tied is only compared to MN with diagonal row and column covariances. Better make this point clear.

2. Figure 4 should show running time instead of training step. I don't think low-rank approximation is gonna influence the convergence that much. It should affect the evaluation speed of each step.

I think the trick the paper uses is a practical one but not significantly novel enough for the ICLR community. It feels like a standard trick people would do when fitting parameters for large matrices, i.e. exploring the low-rank structure and fitting the factorized matrices. Matrix normal is more significant since it reduces the number of parameters as well as maintaining a full covariance matrix with structures. If just focusing on the diagonal covariance, it already throws away the full covariance. Low-rank won't help much with improving the posterior distribution.

**Experience Assessment:**

I have published in this field for several years.

**Review Assessment: Checking Correctness Of Derivations And Theory:**

I carefully checked the derivations and theory.

**Review Assessment: Checking Correctness Of Experiments:**

I assessed the sensibility of the experiments.

**Review Assessment: Thoroughness In Paper Reading:**

I read the paper at least twice and used my best judgement in assessing the paper.

---

> ### Author Response · Authors · 2019-11-15
> **Response to AnonReviewer1**
>
> [R1.1]
> The paper proposes a low-rank approximation to the diagonal of Gaussian mean field posterior which reduces the number of parameters to fit. They show that the predictive performance doesn't drop much compared with the full covariance but the number of parameters is significantly reduced.
>
> [R1.1 our response]
> To be precise we are not comparing our method to the full covariance, which would not be tractable in our settings (the full covariance scales quadratically with respect to the total number of network parameters). Instead, we are comparing our method to the full parametrization of the diagonal covariance, i.e., the standard mean-field approximation. Our method is a further factorization of the diagonal covariance in the parameter space. See Figure 6 for a visual representation of this explanation.
>
> [R1.2]
> Why Matrix normal distribution is related to k-tied Normal distribution when k=1. When k=1, the rank of UV^\top is 1. The covariance is matrix normal is U\otimes V, whose rank is rank(U)rank(V). Also MN has a full covariance for the Gaussian approximation, not mean field. MN is only equal to 1-tied when only diagonal row and column covariances are considered. If that's what the paper means, k-tied is only compared to MN with diagonal row and column covariances. Better make this point clear.
>
> [R1.2 our response]
> We thank the reviewer for the comment. We changed the sentence below from our paper as suggested by the reviewer to make the comparison with the Matrix normal distribution clearer:
> "<MN distribution> is related to our k-tied Normal distribution when k = 1" -> "<MN distribution> is related to our k-tied Normal distribution with $k=1$ when MN uses diagonal row and column covariances."
> We have also added a figure (see Figure 6) to clarify this explanation.
>
> [R1.3]
> Figure 4 should show running time instead of training step. I don't think low-rank approximation is gonna influence the convergence that much. It should affect the evaluation speed of each step.
>
> [R1.3 our response]
> We measured the evaluation speed of each step for a simple model with 2 dense layers [0]. For this model, the measurements show that the k-tied Normal posterior does not decrease the evaluation speed when compared to the standard parameterization of the GMFVI. This experiment can be replicated using the Colab notebook in [1]. We updated the end of section 3.2 and Appendix D to include these results.
>
> More generally, we do not expect a significant change in the evaluation speed of each step when using the k-tied Normal posterior. The biggest additional operation per step when using the k-tied Normal posterior compared to the standard parameterization is the UV^T multiplication to materialize the posterior standard deviations matrix A, where U \in R^{m \times k}, V \in R^{m \times k} and A \in R^{m \times n}. The time complexity of this operations is O(kmn). This time complexity is usually negligible when compared to the complexity of data-weight matrix multiplication with time complexity of O(bmn), where b is the batch size.
>
> [0] https://www.tensorflow.org/tutorials/keras/classification
> [1] https://colab.research.google.com/drive/14pqe_VG5s49xlcXB-Jf8S9GoTFyjv4OF
>
> [R1.4]
> I think the trick the paper uses is a practical one but not significantly novel enough for the ICLR community. It feels like a standard trick people would do when fitting parameters for large matrices, i.e. exploring the low-rank structure and fitting the factorized matrices. Matrix normal is more significant since it reduces the number of parameters as well as maintaining a full covariance matrix with structures. If just focusing on the diagonal covariance, it already throws away the full covariance. Low-rank won't help much with improving the posterior distribution.
>
> [R1.4 our response]
> We acknowledge that MN covers more expressive posterior distribution. However, the goal of our paper is not to investigate richer posterior distributions, beyond mean-field. Instead, we start from the observation that scaling up mean-field and making it more robust is still a challenging problem ([0]). We try to tackle this problem by reducing the number of parameters to optimize---while maintaining a comparable predictive behavior---and obtaining less noisy gradient estimates.
>
> [0] Practical Deep Learning with Bayesian Principles

---

### Decision · Program_Chairs · 2019-12-19

**Decision:**

Reject

**Comment:**

This paper proposes to reduce the number of variational parameters for mean-field VI. A low-rank approximation is used for this purpose. Results on a few small problems are reported.

As R3 has pointed out, the main reason to reject this paper is the lack of comparison of uncertainty estimates. I also agree that, recent Adam-like optimizers do use preconditioning that can be interpreted as variances, so it is not clear why reducing this will give better results.

I agree with R2's comments about missing the "point estimate" baseline. Also the reason for rank 1,2,3 giving better accuracies is unclear and I think the reasons provided by the authors is speculative.

I do believe that reducing the parameterization is a reasonable idea and could be useful. But it is not clear if the proposal of this paper is the right one. Due to this reason, I recommend to reject this paper. However, I highly encourage the authors to improve their paper taking these points into account.